# Universality and quantum criticality in quasiperiodic spin chains

Utkarsh Agrawal [1✉], Sarang Gopalakrishnan[2,3] & Romain Vasseur[1]

Quasiperiodic systems are aperiodic but deterministic, so their critical behavior differs from that of clean systems and disordered ones as well. Quasiperiodic criticality was previously understood only in the special limit where the couplings follow discrete quasiperiodic sequences. Here we consider generic quasiperiodic modulations; we find, remarkably, that for a wide class of spin chains, generic quasiperiodic modulations flow to discrete sequences under a real-space renormalization-group transformation. These discrete sequences are therefore fixed points of a functional renormalization group. This observation allows for an asymptotically exact treatment of the critical points. We use this approach to analyze the quasiperiodic Heisenberg, Ising, and Potts spin chains, as well as a phenomenological model for the quasiperiodic many-body localization transition.

[1] Department of Physics, University of Massachusetts, Amherst, Massachusetts 01003, USA. [2] Department of Physics and Astronomy, CUNY College of Staten Island, Staten Island, New York 10314, USA. [3] Physics Program and Initiative for the Theoretical Sciences, The Graduate Center, CUNY, New York, New York 10016, USA. ✉email: uagrawal@umass.edu

Quenched randomness has dramatic effects on the thermodynamics and response of one-dimensional quantum systems. Infinitesimal randomness can completely change the critical properties of quantum phase transitions, if the correlation length exponent $\nu$ violates the Harris criterion $\nu > 2$[1]. This stability criterion follows from requiring that the average detuning $\delta = g - g_c$ away from criticality be much larger than the detuning fluctuations due to randomness, $\sim \xi^{-1/2}$, with $\xi \sim \delta^{-\nu}$ the correlation length. For a wide variety of models of quantum magnetism, the resulting random critical points are of the infinite-randomness type, for which asymptotically exact renormalization-group (RG) methods exist[2–5]. Many properties of these critical points can be understood by central-limiting arguments that rely on the uncorrelated nature of the disorder potential. Such infinite-randomness fixed points also seem to exist in two dimensions[6].

Many systems of current interest—such as quasicrystals[7–9], twisted bilayer graphene[10–12], and cold atoms in bichromatic laser potentials[13–17]—are inhomogeneous, but with quasiperiodic rather than random modulation of the couplings. Quasiperiodic patterns are deterministic and have long-range spatial correlations: thus the central-limiting arguments that describe random critical points fail in the quasiperiodic case. Clean critical points are more stable to quasiperiodic than to random potentials: the most general stability criterion, due to Luck, is $\nu \geq 1$ (in one dimension)[18]. This generalizes the Harris criterion to the case where fluctuations of quasiperiodic potentials cause $\sim 1/L$ shifts to the detuning over a distance $L$ (in the case of zero wandering constant), instead of the central-limiting behavior $\sim 1/\sqrt{L}$ of the random case. The Heisenberg and Potts models violate even this weaker criterion, so when their couplings are quasiperiodically modulated they flow to a critical point dominated by the modulation. The study of such quasiperiodic quantum critical points is in its infancy. Unlike random couplings, which are typically drawn independently from some distribution, quasiperiodic couplings vary as some periodic function $f(x)$ of space; we do not yet understand which properties of quasiperiodic critical points depend on the nature of this function but it is clear from single-particle hopping problems that smooth, sharp, and singular variation give rise to different phases and critical points. RG methods so far have been restricted to the simplest type of function, a binary substitution sequence[19–27]. Criticality for generic quasiperiodic modulation has only been addressed very recently for free fermions[28–31].

The central result of this work is that, for a wide class of models, generic quasiperiodic modulations flow under renormalization to discrete substitution sequences. These sequences are attractors in the space of quasiperiodic modulations, i.e., fixed points of a functional RG. As generic patterns flow to substitution sequences, for which asymptotically exact real-space RG schemes exist, we can construct asymptotically exact descriptions of generic quasiperiodic quantum critical points, and describe many of their properties (such as critical exponents) analytically. In what follows, we explain why (and under what conditions) substitution sequences are attractors, using the illustrative example of the Heisenberg spin chain; we then extend our analysis to the Ising and Potts models, and finally to a toy model of the many-body localization (MBL) transition[32,33].

## Results

### Quasiperiodic Heisenberg chain.
Although our results are very general and apply to a variety of one-dimensional systems, for concreteness we will illustrate the approach on a paradigmatic example of quantum magnetism: the antiferromagnetic spin-$\frac{1}{2}$ Heisenberg spin chain

$$H = \sum_i J_i \vec{S}_i \cdot \vec{S}_{i+1} \qquad (1)$$

with $J_i > 0$. In the clean case $J_i = J$, this spin chain is gapless and is described at low energy by a $SU(2)$ symmetric Luttinger liquid with Luttinger parameter $g = \frac{1}{2}$. Disorder in the $J_i$ couplings is a relevant perturbation[34] that leads to a quantum critical random-singlet state[2]. In the random case, the low-energy physics can be captured by a strong disorder real-space RG with the following iterative procedure[35,36]: one identifies the strongest remaining coupling $J_i$, forms a singlet out of the spins $i, i+1$ and generates a new effective coupling $J_{eff} = \frac{J_{i-1}J_{i+1}}{2J_i}$—at second order in perturbation theory—between spins $i-1$ and $i+2$. This procedure is accurate so long as $J_i \gg J_{i-1}, J_{i+1}$; in the random case, the ratio of neighboring couplings flows to infinity, so the procedure is asymptotically exact[3].

Here we are interested instead in quasiperiodic modulations of the couplings, with $J_i = f(i)$ with $f(x) = f(x + \varphi^{-1})$ for irrational $\varphi$ [in the bulk of this text we take $\varphi = \frac{1+\sqrt{5}}{2}$] (see, e.g., refs. [37–39]). We take $f > 0$ to be a general smooth function with a smooth logarithm. To understand whether this perturbation is relevant at the Heisenberg critical point we recall that the Heisenberg chain is a critical point separating two inequivalent dimerized phases, which occur when the even and odd bonds have different strengths. The correlation length exponent for the dimerization transition is $\nu = 2/3 < 1$; thus, weak quasiperiodicity is relevant[23,40,41] by the Harris–Luck criterion[18] and the system flows to a quasiperiodicity-dominated fixed point.

*Flow to discrete sequences.* We now apply the standard real-space decimation procedure (described above for the random case) to this quasiperiodic model. It is convenient to write the decimation rule as

$$\ell_i' = \ell_{i-1} - \ell_i + \ell_{i+1} + c, \qquad (2)$$

where we have defined $\ell_j = -\ln J_j$, $c = \ln 2$ (for the Heisenberg model), and $\ell_i = \min\{\ell_j\}$. We will be interested in other values of $c$ below, in the context of the Potts model, so we will treat it as a parameter. For simplicity, we consider the following quasiperiodic potential

$$\ell_j = -\ln J_j = a + \cos(2\pi\varphi j + \theta), \qquad (3)$$

with $\varphi$ the golden ratio, $\theta$ is a random phase, and $a$ is an arbitrary constant. However, we emphasize that our conclusions also hold for any sufficiently regular quasiperiodic potential with frequency $\varphi$ (see Supplementary Note 1.C of the Supplementary Discussion).

Many properties of the decimation procedure can be understood analytically. It is helpful to introduce the notion of a local minimum, i.e., a site $j$ such that $\ell_j < \min(\ell_{j-1}, \ell_{j+1})$. Any such coupling will get decimated before its neighbors, so we can decimate them all at once. It is crucial to note that, if we label minima as $B$ and all other sites as $A$, we will arrive at a two-letter Fibonacci sequence defined by the inflation rule $A \rightarrow AB$, $B \rightarrow A$ (this can be checked by inspection, see Supplementary Note 1.A,B of the Supplementary Discussion for a formal proof). This implies that the first couplings to be decimated already follow a Fibonacci sequence. With this crucial observation in mind, we denote the set of minima as $B_0$ and the set of all other couplings as $A_0$. We now decimate all the $B_0$ couplings—we call this a Fibonacci RG step. This gives rise to a new Fibonacci sequence: once again, we can identify the local minima, denote them $B_1$, decimate them, and so on. This allows us to keep track of the renormalized couplings analytically. Remarkably, after a few steps, we find that

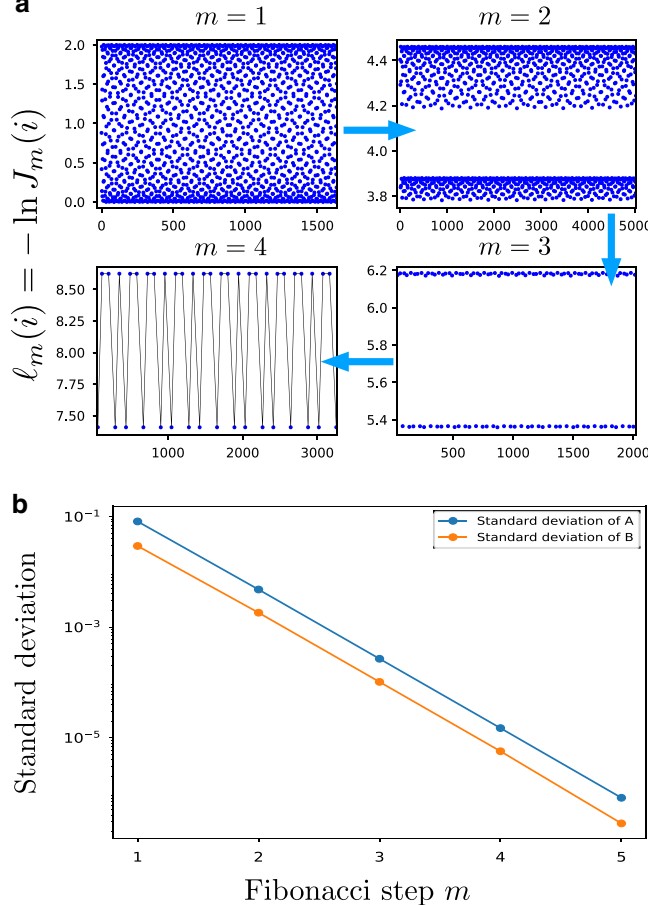

**Fig. 1 Quasiperiodic Heisenberg chain. a** Evolution of the couplings under renormalization for an Heisenberg chain with initial potential (3) with $a = 1$. The fluctuations decay with the number of RG steps and become completely negligible after a few Fibonacci steps. **b** The fluctuations about the sequence prediction (4) starting from a cosine potential decay exponentially with the number of Fibonacci RG steps $m$.

all the $A(i)$ and $B(i)$ at a given step become increasingly similar in magnitude. Specifically, after $m$ Fibonacci RG steps, we find the effective couplings (see Supplementary Note 1.B of the Supplementary Discussion)

$$A_m(i) = a + m(m+1)c + \frac{\cos(F_{3m+2}\pi\varphi)}{\cos\pi\varphi} + \varepsilon_{i,m}^A,$$
$$B_m(i) = a + m^2c - \frac{\cos(F_{3m+1}\pi\varphi)}{\cos\pi\varphi} + \varepsilon_{i,m}^B, \quad (4)$$

where $F_n$ is the $n$th Fibonacci number and $\varepsilon_{i,m}^{A,B}$ are site-dependent fluctuations that go to 0 exponentially as $m \to \infty$. With each Fibonacci step, the fluctuations get smaller and we obtain a sharper sequence that asymptotically becomes a perfect binary Fibonacci sequence. Even if the fluctuations are non-negligible in the initial steps of the RG, they are small enough for us to decimate all $B$ couplings at once (Fig. 1). This means that the complicated initial potential (3) flows under RG to a perfect binary Fibonacci sequence. Moreover, we have $A_m - B_m = mc + o(1)$ as $m \to \infty$, which indicates that the decimation rule (2) given by second-order perturbation theory becomes asymptotically exact as $m \to \infty$, as in the random case. In contrast with the random case, where the fixed point is specified by a probability distribution of couplings, in the quasiperiodic case the fixed point Hamiltonian is specified by a specific, self-similar sequence of

couplings: any sufficiently regular function $f(x) = f(x + \varphi^{-1})$ flows to a binary piecewise function, which is invariant (up to rescaling) under the RG.

We remark that although this flow to nearly discrete sequences is a general property of the RG rules, and occurs for any irrational number, the self-similarity of these discrete sequences is a special property of the Golden Ratio and other irrational numbers with recurring continued fraction expansions, such as the metallic means $1/(n + 1/(n + \ldots))$ (the Golden Ratio is the case $n = 1$). For more general irrational numbers, the sequences do not repeat under the RG: instead, each level of the RG is governed by the coefficient of the continued fraction expansion at the corresponding level[21,28,42,43].

**Quantum critical behavior.** The critical properties of the quasiperiodic chain (1) then follow straightforwardly from Eq. (4), in agreement with previous works on discrete aperiodic sequences. The new fixed point has dynamical exponent $z = \infty$: a chain of length $L \sim \varphi^{3m}$ is fully decimated in $m$ Fibonacci steps, so the energy gap $\Delta E$ of the chain is set by the last coupling to be decimated, $\log \Delta E \sim -m^2 c$, so that $\Delta E \sim e^{-\frac{c}{(3\ln\varphi)^2}\ln^2 L}$[25], where we have used (4). The correlation length exponent $\nu$ can be calculated by introducing an asymmetry between even and odd couplings at the new critical point. Let us assume a perfect sequence of $A_0$ and $B_0$ which is dimerized such that $A_0^E = A_0 + \delta/2$, $B_0^E = B_0 + \delta/2$, $A_0^O = A_0 - \delta/2$, $B_0^O = B_0 - \delta/2$. It is straightforward to check that if one constructs an effective bond made of $n$ microscopic bonds, its coupling will have an additive piece $\pm n\delta/2$ where the sign depends on whether the effective bond is even or odd. Thus, after the RG is iterated out to a scale $L$, the effective couplings at that scale consist of $O(L)$ microscopic couplings, so the asymmetry between the even and odd couplings is $\sim\delta L$. The asymmetry thus becomes of order unity when $L \sim 1/\delta$ and it follows that $\nu = 1$ (see Supplementary Note 1.D of the Supplementary Discussion).

**Quantum Potts model.** We now turn to the $q$-state quantum Potts model, governed by the Hamiltonian

$$H = -\sum_i J_i \delta_{n_i,n_{i+1}} - \sum_i \frac{h_i}{q} \sum_{n_i,n_{i'}} |n_i\rangle\langle n_i'|, \quad (5)$$

where $n_i$ is a variable on site $i$ that takes one of $q$ possible values. To treat this model in the RG scheme, one formally rewrites it as a chain with twice the number of links, and assigns the variables $J_i$ to even links and $h_i$ to odd links. The decimation step[44] then takes the same form as Eq. (2) with $c = \log(q/2)$. More precisely, decimating a strong field $h_i$ leads to a new effective bond $J_{eff} = 2J_{i-1}J_i/(qh_i)$ connecting neighboring spins, while decimating a strong bond $J_i$ creates an effective spin acted on by am effective field $h_{eff} = 2h_i h_{i+1}/(qJ_i)$. When $q > 2$, $c > 0$; thus, once again the RG flows to discrete sequences. The main distinction between the Potts and Heisenberg models lies in the choice of initial couplings: in the Heisenberg model, it was natural to draw all bonds from the same quasiperiodic sequence, whereas here it is natural to take the $h_i$ and $J_i$ from distinct quasiperiodic sequences with frequency $\varphi$ but different phases: $J_i = W_J(a + \cos(2\pi\varphi(i + \frac{1}{2}) + \theta_J))$ and $h_i = W_h(a + \cos(2\pi\varphi i + \theta_h))$ with $a > 1$. This introduces a separate variable to the problem, viz. the relative phase $\theta \equiv \theta_J - \theta_h$ between the sequences for $h_i$ and $J_i$.

Numerically running the RG in this case leads to the following picture. When $\theta$ is close to the special values, we once again observe a flow to self-similar Fibonacci-like sequences. In that case, the results for the XXX spin chain carry over to Potts immediately—in particular, $\nu = 1$. Other critical exponents can

**a**

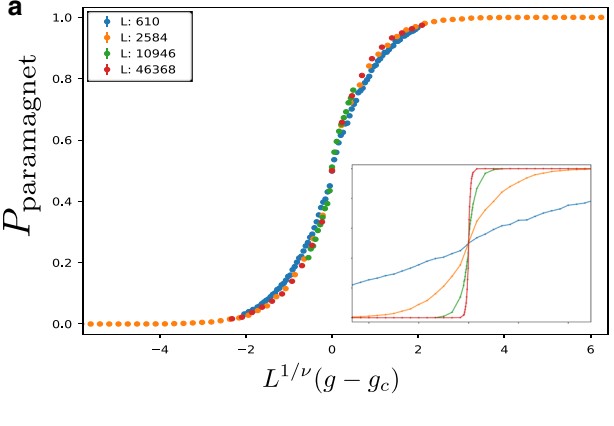

**b**                    **c**

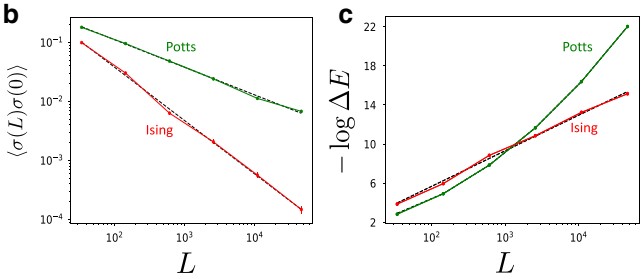

**Fig. 2 Quasiperiodic Potts ($q = 3$) and Ising ($q = 2$) chains.** For the Ising chain we choose a potential with both positive and negative couplings, whereas for the Potts chain all couplings are taken to be antiferromagnetic. **a** Scaling collapse of the probability of the RG to end in a paramagnetic phase for the Potts model, with $\nu = 1$. Here, $g = W_h/W_J$ is an asymmetry parameter between $h_i$ and $J_i$ with $W_{J,h}$ the amplitude of the quasiperiodic potentials and $g_c = 1$. Inset: Raw, uncollapsed data. **b** Spin–spin correlation function $\langle \sigma(L)\sigma(0) \rangle$ averaged over the uncorrelated phases $\theta_J, \theta_h$, scaling as $L^{-0.47}$ for Potts (in good agreement with (6) derived for discrete Fibonacci sequences) and $L^{-0.9}$ for Ising. Error bars represent SE. **c** Energy-length scaling: $\Delta E \sim L^{-0.22 \ln L}$ for Potts, whereas the Ising transition has a finite dynamical exponent $z \approx 1.6$.

readily be computed analytically; e.g., we find that correlation function of the order parameter $\sigma_i^{(a)} = \delta_{n_i,a} - q^{-1}$ with $a$ a given Potts color, scales as (see Supplementary Note 1.E of the Supplementary Discussion)

$$\left\langle \sigma_0^{(a)} \sigma_r^{(a)} \right\rangle \sim r^{-2\Delta_\sigma}, \text{ with } \Delta_\sigma = \frac{\ln(1 + 2\varphi^{-1}/3)}{3 \ln \varphi}. \qquad (6)$$

However, for large $\theta$ we see abrupt transitions to different sequences. Thus, there appear to be multiple fixed-point sequences, with transitions between them occurring at special values of $\theta$. These fixed points have different length-energy scaling: in all cases, $\Delta E \sim e^{-a \ln^2 L}$, but $a$ depends on $\theta$. However, all of the fixed points agree on the exponent $\nu = 1$ and on the spin–spin correlation function (6) (Fig. 2).

**Ising model**. We briefly remark on the $q = 2$ Potts model, i.e., the Ising model. In this case, $c = 0$ in the decimation rule (2). Thus, the RG does not take arbitrary functions to sequences. Instead, nonsingular sequences of $\ell$ are generically squashed under the RG and become effectively constant after a few steps. Remarkably, this corresponds to the observation that weak quasiperiodic modulations are (marginally) irrelevant at the clean Ising critical point. It is indeed known from free fermion numerics[30,31] that the Ising transition in the presence of any such nonsingular quasiperiodic potentials is governed by the clean Ising conformal field theory, in agreement with the predictions of the RG. To see a

nontrivial transition in this case, one must take singular distributions of $\ell$; one can do this, e.g., by taking $J_i = W_J(a + \cos(2\pi\varphi(i + \frac{1}{2} + \theta_J)))$ and $h_i = W_h(a + \cos(2\pi\varphi i + \theta_h))$, with $0 < a < 1$ so $\ell_{2i} = -\ln|J_i|$, $\ell_{2i+1} = -\ln|h_i|$ is singular. For $a > 1$, quasiperiodic potentials flow to uniform ones under renormalization, indicating a transition in the clean Ising universality class. For $a < 1$ the critical dynamics is strongly modified, because the chain has a finite density of nearly broken links, corresponding to a flow to a quasiperiodic fixed point. This quantitatively reproduces the phase diagram obtained in refs. [30,31]. For $a < 1$, the RG does not lead to perfect sequences, and does not flow to "infinite quasiperiodicity" as the examples described above. This means that the perturbation theory steps do not become asymptotically exact under the RG. This is a physical feature of the transition, as it is known from numerics that it has a finite dynamical exponent[30,31]. (This is similar to the random case where the strong disorder RG is exact only for infinite-randomness critical points which have infinite dynamical exponents.) Although our RG procedure is not fully controlled for this model, running the RG numerically yields a fixed point that is qualitatively similar to the numerically seen one[28,30,31] (Fig. 2): in particular, it has a correlation length exponent $\nu = 1$, and a finite dynamical critical exponent $z \approx 1.6$, which is close to the numerical value $z \approx 2$. The remaining discrepancies are to be expected given that the procedure is not controlled; however, the RG correctly captures the qualitative features of the transition and is the only analytic method able to capture the universality class of this transition.

**Quasiperiodic MBL transition**. The observation that generic quasiperiodic potentials flow to analytically tractable substitution sequences under renormalization has broad implications beyond zero-temperature quantum criticality. In particular, isolated quantum systems can undergo eigenstate phase transitions[45], in which the properties of eigenstates (and correspondingly the nature of the dynamics) changes in a nonanalytic way, as some parameter is tuned. A key example of such a transition is the highly studied many-body localization (MBL)[46–48] transition in one dimension, which separates localized and chaotic dynamical phases. In the random case, the nature of the transition remains debated[49–56], but many RG schemes have been proposed[32,33,57–60], with the most recent works supporting a Kosterlitz–Thouless–like transition[33,61,62].

In comparison with the random case, the physics of MBL in quasiperiodic systems remains largely unexplored. Within the latest RG schemes, the mechanism driving the transition is the nucleation of chaos in rare, anomalously thermal regions[53,63,64]. Quasiperiodic systems seem to lack such regions and might therefore have a qualitatively different MBL transition, but the nature of this transition is not fully understood (but see ref. [56]). It would be natural to adapt previous RG approaches to the present case; however, it is not clear whether these schemes are controlled and there are also two inequivalent possible extensions. Some RG schemes in the random case[58,59] attempt to construct typical individual eigenstates. Deep in the localized phase, these look like random product states. The RG hierarchically arranges these product states into resonant clusters. In principle, which spins belong to this resonant cluster depends on which product state one started out in, as nonresonant spins affect the resonance condition for the cluster to form. In random systems, these Hartree-like effects renormalize the initial random potential but do not change its statistical properties. In the quasiperiodic case, however, things are drastically different: a quasiperiodic potential plus random Hartree shifts is in effect a random potential.

Including these random Hartree shifts would thus presumably lead to a transition that looks similar to the random one.

A different class of approaches avoids speaking directly of states, instead trying to disentangle the Hamiltonian by a series of unitary rotations, and thus construct localized integrals of motion (LIOM) in the MBL phase[65–67]. Deep in the MBL phase, a full set of LIOMs exists and, at a critical disorder value, all (or most) of these LIOMs cease to exist. (Note that this is true even if there is a many-body mobility edge, as in this scenario there are some fully thermal eigenstates and the existence of such thermal eigenstates is inconsistent with an extensive number of LIOMs.) There is a transition when the first LIOMs cease to be well-defined; we associate this with the localization transition (note that it must be a sharp transition regardless of whether it is the only localization transition). The properties of the LIOMs in a quasiperiodic system are clearly quasiperiodic, though typical eigenstates have an additional random element from the occupation numbers. Phenomenological RG schemes that model a system in terms of thermal and localized blocks can capture this procedure, as they never explicitly invoke eigenstates.

Motivated by this observation, we now explore the simplest such scheme[32,33]. This model also describes certain classical coarsening dynamics[68], and as we shall see its RG rules are similar to those discussed above. This model assumes that on some coarse-grained scale, the system can be described alternating thermal ($T$) and insulating blocks ($I$), characterized by a single parameter, which we call "length" $\ell_{T,I}$. During the RG, the smallest $I$ or $T$ block is decimated and merged with its neighbors into a new $T/I$ block, in the simplest version of this RG[32], a renormalized length that is the sum of the decimated blocks $\ell_{new}^{T/I} = \ell_{i-1}^{T/I} + \ell_i^{I/T} + \ell_{i+1}^{T/I}$. This RG rule is almost the same as (2), with $c = 0$ and a different sign in front of $\ell_i$. For random initial distributions of the block sizes, this simple model has a second-order phase transition with correlation length exponent $\nu \approx 2.5$. Starting from quasiperiodic initial lengths $\ell_i^{T/I} = W^{T/I}(1 + \cos(2\pi\varphi i + \theta))$, we find that the lengths once again renormalize to a Fibonacci sequence for $W^T = W^I$ (see Supplementary Fig. 2).

We now consider perturbing away from these fixed points, e.g., driving the system into the insulating phase by making the $I$ blocks slightly larger than the $T$ blocks. We can do this with the same dimerization perturbation as we previously had. A key thing to note is that this perturbation is actually irrelevant at the Fibonacci-sequence fixed point: a small perturbation does not change the sequence of decimations and the perturbation strength decreases under decimation, as a consequence of the crucial sign change in the RG rule. This situation changes, however, for continuous initial distributions: here, there are pairs of adjacent couplings that are nearby in energy; thus, there are minima (at the critical point) that cease to be minima when the system is perturbed. Thus, even small perturbations change the order in which sites are decimated and disrupt the fixed-point sequence. In this sense, dimerization is a dangerous irrelevant perturbation at the quasiperiodic MBL fixed point.

Interestingly, regardless of this subtlety, we once again find a correlation length exponent $\nu = 1$ for smoothly varying initial conditions. The argument is as follows: define a defect as a bond that was a local minimum for the critical pattern but ceased to be one after being perturbed (or vice versa). If a defect is present at position $i = 0$, then another defect will be created at $i = F_n$ with $n$ large enough, as $\varphi F_n$ is almost an integer. Going away from criticality by an amount $\delta W$ induces defects if we choose $n$ so that the deviation of $F_n$ from being the exact period (given by the fractional part of $\varphi F_n$) satifies $\{\varphi F_n\} = \varphi^{-n} \sim \delta W$. The distance between the defects then sets the correlation length $\xi \sim F_n \approx \varphi^n \sim \delta W^{-1}$, implying that $\nu = 1$ (see Supplementary Note 2.A of

the Supplementary Discussion and Supplementary Figs. 3, 4). An interesting consequence is that one can tunably vary $\nu$ by introducing singularities at appropriate points in the original sequence; we have checked that this is in fact true (see Supplementary Note 2.B of the Supplementary Discussion and Supplementary Fig. 5). This general argument applies to all the transitions studied in this paper and does not rely on the precise nature of the underlying fixed points.

This toy RG is perfectly symmetric between the $T$ and $I$ phases; in realistic models, this symmetry is absent, as thermal blocks are much more infectious than insulating blocks. To introduce some asymmetry between the two phases, we follow ref. [33] and introduce a new parameter $\beta$ in the RG rules

$$\ell_{new}^{T/I} = \ell_{i-1}^{T/I} + \beta^{I/T}\ell_i^{I/T} + \ell_{i+1}^{T/I}, \tag{7}$$

with $\beta^I = 1/\beta^T = \beta$. This modified RG is analytically solvable for random distributions and captures much of the phenomenology of the MBL transition in the maximally asymmetric limit $\beta \to \infty$, with a KT-type transition ($\nu = \infty$) in the random case. In the quasiperiodic case, weak asymmetry is irrelevant at the fixed point; however, as the asymmetry is cranked up, eventually the order of decimations changes and one flows to a new fixed-point sequence. Thus, as $\beta$ is increased, the system discontinuously jumps between a number of increasingly complicated self-similar fixed-point sequences, which we detail in Supplementary Note 2. C of the Supplemental Discussion. Perturbations away from criticality lead to defects in these self-similar sequences, with the same exponent $\nu = 1$ from the argument above. We therefore conjecture that the value $\nu = 1$ remains exact as $\beta \to \infty$. Extrapolating our results from this phenomological RG to the quasiperiodic MBL transition leads to the natural prediction that it belongs to a universality class different from the random case, as conjectured in ref. [55]. Moreover, our approach leads to the novel prediction that $\nu = 1$, in agreement with recent exact diagonalization results[55,69]. It would be interesting to investigate the consequences of the dangerously irrelevant character of the parameter driving the transition in future work.

## Discussion

To summarize, we have shown that under quite general conditions, conventional real-space RG rules attract generic initial conditions to fixed-point sequences that are self-similar under RG. The simplest case is in the Heisenberg spin chain, where the fixed-point sequence is given by the simple Fibonacci deflation rule; many results were previously known for this sequence, but our work shows that these results apply far more generally than had been appreciated. For the Potts model, we again found attracting sequences, but unlike the Heisenberg model we found multiple distinct attractors. Finally, for the toy MBL transition models, we found a much more unexpected type of RG flow, in which the control parameter driving the transition is in fact dangerously irrelevant at the critical point, but nevertheless the exponent $\nu = 1$. It would be interesting to investigate whether our results can be extended to two-dimensional quasiperiodic systems[70].

## Methods
**Numerical implementation.** Numerical results were obtained by running the real-space RG procedure iteratively, by identifying strong couplings in the Hamiltonian and decimating them before weaker ones, putting the spins involved in a strong coupling in their local ground state. Residual couplings are then computed using second-order perturbation theory. For all quasiperiodic systems discussed in this paper (with the notable exception of the quantum Ising chain), we find that the small parameter controlling the validity of the perturbation theory decreases under RG, making the procedure asymptotically exact. For quasiperiodic potentials of the form $\cos(2\pi\varphi j + \theta)$, we average our results numerically over thousands of choices of the global phase $\theta$.

## Data availability
Data are available on request from the authors.

## Code availability
Codes are available on request from the authors.

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

## Acknowledgements

We thank A. Chandran, P. Crowley, D. Huse, V. Khemani, and C. Laumann for stimulating discussions. We also thank S. Gazit, B. Kang, and J. Pixley for useful discussions and for sharing with us their unpublished Quantum Monte Carlo results. This work was supported by the US Department of Energy, Office of Science, Basic Energy Sciences, under Early Career Award No. DE-SC0019168 (U.A. and R.V.), the Alfred P. Sloan Foundation through a Sloan Research Fellowship (R.V.), and by NSF Grant Number DMR-1653271 (S.G.).

## Author contributions

U.A., S.G., and R.V. designed and performed research, analyzed data, and wrote the paper.

## Competing interests

The authors declare no competing interests.
