## [Peer Review File · Nature Communications]

Reviewers' comments:

Reviewer #1 (Remarks to the Author):

The manuscript deals with universality and quantum criticality in quasiperiodic spin chains. The topic is of a considerable interest specially in connection with many-body-localized phases which are expected to be different for quasiperiodic modulations. Nevertheless the manuscript cannot be accepted in its present form as the major claims are not clearly sustained and need more evidence.

The authors consider generic quasiperiodic modulations of a wide class of spin-chains and show that quasiperiodic modulation flows to discrete sequences under a real-space renormalization group transformation. These discrete sequences are therefore fixed points of a functional renormalization group. They claim that this observation allows for an asymptotically exact treatment of the critical points.

-The claim of the authors is not convincing since the real-space decimation rule they apply depends strongly on the value of "c" which is $\ln(2)$ for the Heisenberg model and is assumed to be a parameter for Potts model. The flows to a discrete sequence works well only for $c > 0$. In fact when c is zero the RG does not take arbitrary functions to sequences and the RG becomes uncontrolled. The RG becomes only qualitative in this case, as they have found for the Ising model. Thus the initial claim fails and the result is not generic.

-As for the toy MBL transition models the result they find looks really strange as they claim that a dangerously irrelevant control parameter leaves the critical exponent unchanged i.e. $\nu = 1$, at odds with the KT transition for a random distribution. One would naively expect a critical parameter β_c that drives the system to a KT transition.

It's not even clear if they find fixed-point sequences.

-The bibliography is reach but they don't quote the paper:

PHYSICAL REVIEW LETTERS 123, 070405 (2019)

Critical Behavior and Fractality in Shallow One-Dimensional Quasiperiodic Potentials by Heping Yao, Hakim Khoufli, Lea Bresque, and Laurent Sanchez-Palencia. It would be useful to compare to the results in this paper where the critical exponent is found $\nu = 1/3$, less than one.

Minor points:

-They quote at least two criteria for the correlation functions exponents in the text and it would be good to briefly describe them not pretending the reader to be aware of them

-In fig. 1 they say that the fluctuations become completely negligible after only 3 Fibonacci steps. They seem of order 1, so the statement is not clear, they should better describe the results of the figure.

Reviewer #2 (Remarks to the Author):

In this manuscript, the authors study the emerged quantum criticality of spin chains subjected to a quasi-periodic modulation, which is a growing and trending topic. They apply the strong disorder real-space RG scheme to the quasi-periodic coupling strength, mainly focusing on the golden ratio as the irrational period. In the simplest case of the quantum Heisenberg model, they find analytically that a general smooth quasi-periodic coupling flows to the discrete binary Fibonacci sequence, and that the model becomes self-similar after a few RG step. Based on the result, many properties of this critical point dominated by the quasi-periodic modulation can be deduced. They also note that the flow to the discrete sequence to occurs for general irrational periods. Based on this analytical understanding, they also study the quantum Potts model, which exhibits richer

physics. Generalization of the RG equation can also be applied to a toy model of the MBL transition in quasi-periodic systems beyond low-lying energy physics.

In my opinion, this work includes a nice set of results, which can be applied to a wide class of systems with quasi-periodic potential and provide an analytical handle to understand some of their properties. This work is of broad interest to physicists in both condensed matter and cold-atom community. Therefore, I am happy to recommend its publication in Nature Communication. Some minor suggestions are listed below.

1) The discussion in the paragraph starting with "many properties of the decimation procedure..." on page 2 should be expanded since it is the main message of the paper. It is difficult to get the key idea without referring to the appendix. The authors should at least write down the RG rules of A and B, which is hidden in appendix I. B. Also, in appendix I. A, the sentence "Denote the corresponding values of the position i as x and y " is difficult to understand.

2) Please define g in Fig 2. (a).

3) It would be nice to see the decimations rule of h and J written down in the context of the discussion on the Potts model.

4) It seems that new fixed points with the same $\nu=1$ appear in both the quantum Potts model and the toy model of the MBL transition when the Fibonacci RG equation is significantly perturbed. Is there a physical understanding of these new fixed points? What are their implications on the models?

Response to Reviewer #1

We thank the referee for their report, and we are glad to see that the referee considers our results to be of considerable interest. The main objection to publication seems to be of technical nature, and we believe that it stems from a point that we now realize was unclear in our manuscript. Namely, as explained in more detail below, the fact that our RG doesn't flow to sequences for the Ising model is an important physical consequence of quasiperiodicity being irrelevant at the clean Ising fixed point, and not a drawback of the method. We have significantly expanded the section on the Ising model to emphasize the accuracy of the RG's predictions. Similarly, our main prediction for the MBL transition is not "strange" and is in fact consistent with earlier proposals.

We have rewritten and expanded the corresponding sections, and believe that these points are much clearer now. Given this, we hope that the referee will also recommend publication in Nature Communications.

“-The claim of the authors is not convincing since the real-space decimation rule they apply depends strongly on the value of "c" which is $\ln(2)$ for the Heisenberg model and is assumed to be a parameter for Potts model. The flows to a discrete sequence works well only for $c > 0$. In fact when c is zero the RG does not take arbitrary functions to sequences and the RG becomes uncontrolled. The RG becomes only qualitative in this case, as they have found for the Ising model. Thus the initial claim fails and the result is not generic. “

We thank the referee for this comment, which made us realize that the section on the Ising model was too concise. The fact that the RG doesn't flow to sequences for the Ising model is absolutely not a pitfall of the method, but rather a very physical feature. Weak quasiperiodic modulations are known to be (marginally) irrelevant at the Ising critical point. Our RG is designed to deal with strong quasiperiodic modulations (just like it is designed to describe strong and infinite-randomness critical points in the random case): it is therefore remarkable that this approach is able to (correctly!) predict the irrelevance of quasiperiodicity. In fact, the RG also predicts quantitatively the phase diagram of the quasiperiodic Ising model: for couplings chosen as $a + \cos(2\pi\phi_i + \theta_{J,h})$, the RG flows to uniform couplings for $a > 1$, while it gives a non-trivial quasiperiodic critical behavior for $0 < a < 1$. This is consistent with the phase diagram of the recent references 30 and 31 that use free fermion numerics. For the critical regime $0 < a < 1$, the RG is indeed not fully exact but this was expected since the corresponding critical point is not of "infinite-quasiperiodicity" type, and has a finite dynamical exponent. (This is again precisely the same as in the random case: the strong disorder RG gives exact predictions for infinite randomness critical points, and is not fully controlled for finite randomness transitions.) Moreover, we emphasize that the RG nevertheless yields predictions that are in remarkable agreement with Refs 30,31, and as far as we know of, this is the only analytical technique able to describe such transitions.

To sum up, while the RG doesn't yield infinite-quasiperiodicity sequence fixed points for the Ising model, this is a feature of the model and not of the RG. If anything, the RG still remains remarkably accurate in its predictions in this case. We have significantly expanded the section on the Ising model to clarify this.

“-As for the toy MBL transition models the result they find looks really strange as they claim that a dangerously irrelevant control parameter leaves the critical exponent unchanged i.e. $\nu=1$, at odds with the KT transition for a random distribution. One would naively expect a critical parameter β_c that drives the system to a KT transition. It's not even clear if they find fixed-point sequences.”

While this is still debated, the natural expectation in the community is that the MBL transitions in the random and quasiperiodic cases are in different universality classes. (see Ref 54). The fact that our quasiperiodic RG doesn't predict a KT transition is therefore not “strange” at all, and in fact our prediction for the critical exponent $\nu=1$ agrees with exact diagonalization studies. We do indeed predict a more surprising dangerously irrelevant perturbation driving the transition, but beyond this, our predictions for the quasiperiodic transition are in fact quite natural from general grounds and previous studies. We have added sentences to clarify this issue.

“-The bibliography is reach but they don't quote the paper:
PHYSICAL REVIEW LETTERS 123, 070405 (2019)
Critical Behavior and Fractality in Shallow One-Dimensional Quasiperiodic Potentials by
Hepeng Yao, Hakim Khoudli, Lea Bresque, and Laurent Sanchez-Palencia. It would be useful to compare to the results in this paper where the critical exponent is found $\nu=1/3$, less than one. “

We thank the referee for pointing out this reference that we included in our bibliography. However, the exponent ν is not directly comparable to our exponents probably because of the definition used in this reference which relies on single-particle properties and involve the fractality of the single-particle wave-functions. In fact, from the Harris-Luck criterion the correlation length exponent cannot be smaller than 1.

“-They quote at least two criteria for the correlation functions exponents in the text and it would be good to briefly describe them not pretending the reader to be aware of them”

We have added a brief description of the Harris and Harris-Luck criterions.

“-In fig. 1 they say that the fluctuations become completely negligible after only 3 Fibonacci steps. They seem of order 1, so the statement is not clear, they should better describe the results of the figure.”

The precise statement is that fluctuations decay exponentially with the number of Fibonacci steps. In practice, this means that they become almost invisible at the scale of the figure after a few steps. We have clarified this point.

Response to Reviewer #2

We thank the referee for reviewing our work and for enthusiastically recommending publication in Nature Communications. We were happy to follow the referee's suggestions, that we address in more detail below.

“1) The discussion in the paragraph starting with “many properties of the decimation procedure...” on page 2 should be expanded since it is the main message of the paper. It is difficult to get the key idea without referring to the appendix. The authors should at least write down the RG rules of A and B, which is hidden in appendix I. B. Also, in appendix I. A, the sentence “Denote the corresponding values of the position i as x and y ” is difficult to understand. “

We have expanded this section to emphasize the main message. We have chosen to keep some technical details in appendix to avoid cumbersome equations in the main text, but we hope the discussion is now clearer. We have also clarified the sentence in the appendix.

“2) Please define g in Fig 2. (a).”

Fixed.

“3) It would be nice to see the decimations rule of h and J written down in the context of the discussion on the Potts model.”

We agree. We have added these decimation rules.

“4) It seems that new fixed points with the same $\nu=1$ appear in both the quantum Potts model and the toy model of the MBL transition when the Fibonacci RG equation is significantly perturbed. Is there a physical understanding of these new fixed points? What are their implications on the models?”

The exponent $\nu=1$ is in some way “superuniversal”, as it follows from the very general defect argument given in the main text. For both the toy model of the MBL transition and the Potts model for various values of the parameter θ , the critical points are controlled by more complicated sequences, making a general analytical treatment beyond the scope of our work. However, many properties such as the value of the correlation length exponent or the scaling of the gap in the Potts chain are common to all these fixed points, and can be derived on general grounds. We have added a sentence to emphasize the generality of the defect argument.

REVIEWERS' COMMENTS:

Reviewer #1 (Remarks to the Author):

The authors have significantly expanded the section on the Ising model to emphasize the accuracy of their RG's predictions and answered appropriately to my concerns.

They have clarified, rewritten and expanded the corresponding sections, and I believe that these points are much clearer now. Given this, I can now recommend the manuscript for publication in Nature Communications.

Reviewer #2 (Remarks to the Author):

The accessibility of the manuscript is improved in this revised version. The authors also clarified the RG procedure of the Ising case based on reviewer #1's comment. I'm happy to reiterate my previous assessment to recommend publication in Nature Communication.